# Me-NDT: Neural-backed Decision Tree for visual Explainability of deep Medical models

**Guanghui Fu**[1]                                   FUGUANGHUI@EMAILS.BJUT.EDU.CN

**Ruiqian Wang**[1]                                  WANGRUIQIAN@EMAILS.BJUT.EDU.CN

**Jianqiang Li**[1]                                       LIJIANQIANG@BJUT.EDU.CN

[1] *Beijing University of Technology, China.*

**Maria Vakalopoulou**[2,3]              MARIA.VAKALOPOULOU@CENTRALESUPELEC.FR

[2] *CentraleSupélec, Université Paris Saclay,* [3] *Inria Saclay, France*

**Vicky Kalogeiton**[4]                    VICKY.KALOGEITON@LIX.POLYTECHNIQUE.FR

[4] *LIX, Ecole Polytechnique, IP Paris, France*

## Abstract

Despite the progress of deep learning on medical imaging, there is still not a true understanding of what networks learn and of how decisions are reached. Here, we address this by proposing a Visualized Neural-backed Decision Tree for Medical image analysis, Me-NDT. It is a CNN with a tree-based structure template that allows for both classification and visualization of firing neurons, thus offering interpretability. We also introduce node and path losses that allow Me-NDT to consider the entire path instead of isolated nodes. Our experiments on brain CT and chest radiographs outperform all baselines. Overall, Me-NDT is a lighter, comprehensively explanatory model, of great value for clinical practice.

**Keywords:** Model explainability, Medical image analysis

## 1. Introduction

Deep learning has broad prospects in the medical field, as it is crucial to establish systems that assist diagnosis. Most methods, however, assist without knowing the decision basis (Lucieri et al., 2020). To increase their explainability, in this work, we exploit recent advances in explainable computer vision, which can be grouped into saliency maps and sequential decision processes. Saliency maps explain predictions by identifying pixels that affect the prediction the most. For instance, (Xu et al., 2015) use attention to generate soft and hard saliency maps for visualization. Similar, (Xiao et al., 2015) use attention to increase explainability. However, these typically focus solely on the relationship between input and output, but not on the decision process and hence are not comprehensive. Most sequential processes are modelled with hierarchy, e.g. decision tree. However, their performance is worse than simple deep methods (Rajendran and Madheswaran, 2010). To achieve a balance between accuracy and interpretability, recently (Balestriero, 2017; Yang et al., 2018) combine deep learning and decision trees. For instance, the Neural-backed Decision Trees (NBDTs) (Wan et al., 2020) is a hierarchical classifier that retains interpretability by using sequential discrete decisions. For medical imaging, we want to know both which area is a suspicious lesion and how to make model decisions. Inspired by NBDT, we propose the Visualized **N**eural-backed **D**ecision **T**ree for **Me**dical image analysis (Me-NDT) that visualizes high-response areas to determine the basis for decision-making. We experiment on two medical datasets and outperform all baselines. Me-NDT brings two advantages: (i) adding the tree structure improves the medical image classification performance; (ii) it is explanatory, as it offers visualization of the decision path and basis of the model.

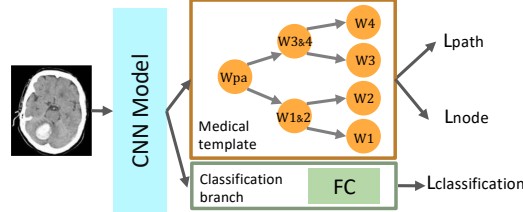

Figure 1: Me-NDT consists of a CNN and a tree structure for medical diseases.

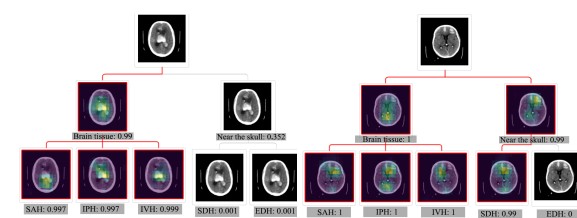

Figure 2: Brain CT visualization results.

## 2. Me-NDT: Visualized Neural-backed Decision Tree for Medical images

Inspired by (Wan et al., 2020), Me-NDT consists of a CNN-basis and two branches, the classification branch and the medical tree-structure template (Figure 1). For CNN-basis we use the convolutional layers of VGG-Net. This is followed by adaptive max pooling that results to a 512 sized feature vector, which is then used as input to the medical template.

**The Medical Tree Template** is constructed with a bottom-up way approach on the anatomical positional relationships and the dependencies among diseases. All diseases are placed at the bottom (children nodes) and then the parent nodes are inferred from the children in a higher level, i.e. as long as one child is true, the parent is also true. The parents are grouped depending on their anatomical appearance. For instance, the parent of SAH, IPH and IVH is defined as brain tissue as it comprises hemorrhagic manifestations in the brain tissue, whereas SDH and EDH are grouped together in the parent node as they appear near the skull. In that way, Me-NDT supervises nodes, and the final prediction is influenced both by each node and its decision path. Note that each node in the tree is represented with a linear layer.

**Losses.** In the medical domain, a image often contains multiple diseases; hence, we cast the problem as a multi-label multi-class classification problem. We train the classification branch with a multi-label cross-entropy loss, i.e. $L_{cls}$. To train the medical template, we introduce two losses: node loss, and path loss. **The *Node loss*** supervises all nodes $n$ (both children $n^{ch}$ and parents $n^{pa}$) with a multi-label cross-entropy: $L_{node} = -\sum_{i=1}^{c}\left((y_i^n \cdot \log(\hat{y}_i^n) + (1-y_i^n) \cdot \log(1-\hat{y}_i^n)\right)$, where $c$ is the number of classes, $y_i^n$ the ground-truth label and $\hat{y}_i^n$ the prediction.

**The *Path loss*** finds the tree path that leads to the predicted disease using multi-label cross-entropy: $L_{path} = -\sum_{i=1}^{c}\left((y_i^n \cdot \log(\hat{y}_i^{n^{pa}} \cdot \hat{y}_i^{n^{ch}}) + (1-y_i^n) \cdot \log(1-\hat{y}_i^{n^{pa}} \cdot \hat{y}_i^{n^{ch}})\right)$, where $\hat{y}_i^{n^{pa}}$, $\hat{y}_i^{n^{ch}}$ the predictions for the parent and children nodes. Note, this loss considers both parent and children nodes that lead to a result. **The *Overall loss*** is: $L = w_c L_{cls} + w_p L_{path} + w_n L_{node}$. We use $L_{cls}$ to accelerate convergence at training; however, at test time, the final prediction is made solely by $L_{path}$ and $L_{node}$. Thus, the visualized classification results represent the decision path of Me-NDT.

**Visualize Decision Process.** Similar to (Zhou et al., 2016), we perform global max pooling on the each node's feature map and use these as features for a fc layer. We obtain the regions importance by projecting the weights of the nodes feature maps to the image:

we only obtain the three largest nodes. Since the child needs to consider the entire path, the input is not the original feature map, but the parent node weighted feature vector.

## 3. Experiments and Discussion

We conduct experiments on the 2D brain CT[1] dataset using VGG-Net. For acceleration, (a) we pre-train the networks for disease classification and initialize Me-NDT with these weights; and (b) we sub-sample the training set while maintaining their distributions and perform 5-fold cross-validation. We set $(w_c, w_p, w_n)=(1, 1000, 10)$. We report F1-score, Recall and Precision results in Table 1. Me-NDT outperforms all baselines, while requiring fewer parameters, hence less computation. Figure 2 displays the decision process, highlighting that Me-NDT accurately indicates the areas of interest while correctly predicting the disease.

| Model | $L_{\mathrm{cls}}$ | # Parameters | F1-score | Precision | Recall |
|---|---|---|---|---|---|
| VGG | – | 134,281,029 | 0.79686 | 0.80947 | 0.81915 |
| **Me-NDT** | – | **16,557,905** | 0.80349 | 0.81513 | 0.82556 |
| **Me-NDT** | ✓ | 16,560,470 | **0.80535** | **0.81679** | **0.82755** |

Table 1: Comparison of various baselines on brain CT.

**Conclusion.** In this study, we introduced Me-NDT, a simple model capable of classifying diseases and visualizing the decision path, thus offering interpretability, crucial for clinical practices. Me-NDT outperforms all baselines while being lighter and faster. Future work includes constructing free-form trees and large-scale experimentation.

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
