# OpenReview forum: "Me-NDT: Neural-backed Decision Tree for Visual Explainability of Deep Medical Models"
_MIDL.io/2021/Conference/Short — MIDL 2021 Poster_

### Official Review · Reviewer_PaRB · 2021-04-30

**Confidence:** 3
**Final Rating:** 1

**Summary:**


This paper uses a recently proposed approach, the Neural-backed Decision Tree, to improve the performance and interpretability of the deep learning methods. Their approach is similar to Wan et al., ICLR 2021, and appends a decision tree to CNN classifier for multi-class disease classification. The decision tree or the medical template assists CNN's learning by using path and node loss. The proposed model shows improvement in terms of the F1-score and is claimed to be more interpretable. Interpretability is achieved via saliency visualization using an approach similar to Zhou et al., CVPR 2016, and medical template.


**Strengths:**


- This approach tries to tackle interpretability, which is important, and current deep learning solutions suffer at addressing this.
- Using NBDT like approach could be a good way to augment deep learning with domain knowledge and thus useful to improve performance on small datasets.

**Weaknesses:**

- Even though this paper is a step in the right direction, it could have been clearer at various points.
  - It is unclear if the interpretability is due to the decision tree or the saliency maps, or both.
  - It is unclear how complex is the decision tree.
  - Is there a relation between classifier weights and decision tree parameters? How is the decision tree helping the classifier learn better? If there is no connection between parameters, these can produce two different results and hence wrong interpretation.
  - What is the meaning of the phrase ".... at test time, the final prediction is made solely by Lpath and Lnode"? How is loss used for prediction? Did the authors mean to imply that the decision tree paths are considered for prediction?

- The authors mentioned using VGG-Net architecture for this problem. However, the difference between the parameters of Me-NDT and VGG-Net suggests otherwise (Table 1). Mainly why is the difference in parameters so large if they are the same CNN architectures?

- The performance difference between the two models, VGG and Me-NDT, could be because of different architecture than decision trees.

- How was VGG-Net trained (Table 1, first row) if it did not use Lcls?

The above observations decrease my confidence in the paper.


**Deanonymize Review:**

no

**Detailed Comments:**


- What is the exact classification task? Mainly, which diseases were considered? Did the dataset also have healthy patients?
- Why does Me-NDT use ~8x fewer parameters if they are based on the same VGG-Net architecture?
- Since the decision tree is hand-designed, how can it help interpretability? Mainly, by predicting the diseases even with a normal CNN, one could think about the decision path.
- See weaknesses.


**Justification Of The Rating:**

My main concern with this paper is that it is unclear how it improves interpretability. Moreover, the results reported in Table 1, mainly concerning the number of parameters cast suspicion on the reported evaluation (see weaknesses).

The paper would benefit from substantial editing and revisiting the experiments.

**Paper Type:**

validation/application paper

**Special Issue:**

no

---

### Official Review · Reviewer_eu5j · 2021-05-04

**Confidence:** 4
**Final Rating:** 4

**Summary:**

The proposed method takes inspiration from Neural-backed decision trees and extends them by introducing a visualization for high-response areas so as to help determine the basis for decision making. The authors argue their approach both improves classification performance and adds explainability. It is validated on 2D brain CT scans of intracranial haemorrhage against a VGG baseline.


**Strengths:**

- Addresses a key problem within medical applications, that is interpretability.

- The loss function is defined such that it enforces interpretability. The network resembles a multi-layered tree structure where the nodes at the final layer are disease outcomes. In its pure form, the loss is simply a sum of node and path losses (overall classification loss added for speed of convergence). Why do the authors need to add L_cls if the nodes are labelled and already denote the presence of a given disease though?

- Classification performance is incrementally higher than a VGG baseline, but the model is also parameter-efficient and explainable.
The write-up is very clear.


**Weaknesses:**

- Fig. 2 – how is the meaning of parent nodes given? E.g. if the children nodes are diseases, how can it be inferred in a deeper tree that in a given intermediate layer a given node corresponds to “Brain tissue” and another to “Near skull”? It seems giving meaning to the tree layers is essential to achieve real interpretability and it is not mentioned in the paper.


**Deanonymize Review:**

yes

**Detailed Comments:**

- Is it possible to train without L_cls and simply with L_node + L_path?

- If we label all nodes in the tree (i.e. label parents given the children), is it not possible to simply train with a single loss? I.e. the nodes that lie in the path to the true disease should be labelled true, otherwise if they lie on a different path, nodes are labelled false? I believe this would be the same as using only L_node. Did you find that adding L_path is also needed to enforce correct behaviour similarly to L_cls?

**Justification Of The Rating:**

For a short paper, the motivation of the paper is strong and the implementation is interesting. The write-up is also clear. I think given the importance of interpretability and the novelty of the method, it deserves a strong accept.


**Paper Type:**

both

**Special Issue:**

yes

---

### Meta-Review · Program_Chairs · 2021-05-11

**Recommendation:** Accept (Poster)
**Confidence:** 5

**Metareview:**

Two reviewers have different opinions about this work. The weakness of this work is that it lacks some details about the decision tree. However, it is an interesting work that can address a key problem for deep learning for medical applications. In general, I maintain acceptance for this conference to raise discussion in the community.

---

### Decision · Program_Chairs · 2021-05-11

Accept (Poster)